# Whole-Cell Display of Phospholipase D in *Escherichia coli* for High-Efficiency Extracellular Phosphatidylserine Production

**DOI:** 10.3390/biom14040430

**Published:** 2024-04-02

**Authors:** Baotong Sun, Zhongchen Li, Yanhong Peng, Fei Wang, Yibin Cheng, Yang Liu, Lixin Ma

**Affiliations:** State Key Laboratory of Biocatalysis and Enzyme Engineering, Hubei Key Laboratory of Industrial Biotechnology, School of Life Sciences, Hubei University, Wuhan 430062, China; 15972990549@163.com (B.S.); lizc0408@163.com (Z.L.); pyhong@stu.hubu.edu.cn (Y.P.); wangfei@hubu.edu.cn (F.W.); chengyibin@hubu.edu.cn (Y.C.)

**Keywords:** phosphatidylserine production, visible sfGFP, phospholipase D, cell-surface display, reused

## Abstract

Phospholipids are widely utilized in various industries, including food, medicine, and cosmetics, due to their unique chemical properties and healthcare benefits. Phospholipase D (PLD) plays a crucial role in the biotransformation of phospholipids. Here, we have constructed a super-folder green fluorescent protein (sfGFP)-based phospholipase D (PLD) expression and surface-display system in *Escherichia coli*, enabling the surface display of sfGFP-PLDr34 on the bacteria. The displayed sfGFP-PLDr34 showed maximum enzymatic activity at pH 5.0 and 45 °C. The optimum Ca^2+^ concentrations for the transphosphatidylation activity and hydrolysis activity are 100 mM and 10 mM, respectively. The use of displayed sfGFP-PLDr34 for the conversion of phosphatidylcholine (PC) and L-serine to phosphatidylserine (PS) showed that nearly all the PC was converted into PS at the optimum conditions. The displayed enzyme can be reused for up to three rounds while still producing detectable levels of PS. Thus, *Escherichia coli*/sfGFP-PLD shows potential for the feasible industrial-scale production of PS. Moreover, this system is particularly valuable for quickly screening higher-activity PLDs. The fluorescence of sfGFP can indicate the expression level of the fused PLD and changes that occur during reuse.

## 1. Introduction

Phosphatidylserine (PS) is an important component of the cell membrane in eukaryotic cells. It is the most abundant anionic phospholipid in eukaryotic organisms, accounting for 10% of the total cellular lipids. PS is widely used in the food, health product, chemical, and pharmaceutical industries. PS plays a significant role in cell apoptosis and blood clotting [1]. PS is essential for healthy neuronal membranes and myelin sheaths [2] and acts as a global immune suppressor in phagocytosis, infectious diseases, and cancer [3]. PS can enhance learning and memory abilities and has been studied for its potential in preventing Alzheimer’s disease [4,5].

Traditionally, PS is extracted from soybeans, vegetable oil, egg yolk, and biomass. However, its low availability and high extraction cost are limiting factors [6,7]. Phospholipase D (PLD, EC. 3.1.4.4)-mediated phosphatidylcholine (PC) transphosphatidylation with L-serine is a promising method for synthesizing PS. Compared to chemical synthesis and extraction methods, PLD enzyme synthesis of PS has the advantages of mild reaction conditions and minimal formation of byproducts [8]. PLD has phospholipidation activity and can be used for the enzyme-catalyzed synthesis of various phospholipids. There have been many reports on the mediated synthesis of natural and specially designed PLs with functional head groups by readily available phosphatidylcholine or phospholipids using PLD [9]. PLD also has hydrolytic activity, converting PC into the unwanted byproduct phosphatidic acid (PA). In recent years, various novel phospholipase D (PLD) enzymes with higher phosphatidylcholine esterification activity and lower hydrolytic activity have been obtained through gene-mining strategies and protein engineering. Among them, PLDs from *Streptomyces* species are more suitable for the efficient synthesis of phospholipids and phospholipid derivatives because they possess higher phospholipid acylation activity compared to PLDs from plants and fungi [10,11,12].

The production of PLD in *Streptomyces* is very low. Therefore, people have already heterologously expressed PLD from different sources in *Escherichia coli* [13], *Pichia pastoris* [14], and *Bacillus subtilis* [15], resulting in a slight increase in activity. In these heterologous expression systems, *Escherichia coli* is the main host for heterologous phospholipase expression, accounting for approximately 86% of all hosts [16]. However, most reports on the heterologous expression of PLD in *Escherichia coli* are related to intracellular expression, which is disadvantageous for enzyme isolation and industrial production. Additionally, it has been reported that the accumulation of PLD in cells can lead to plasmid instability, cell lysis, and a further decrease in PLD activity [13,17]. Cell-surface display offers a more cost-effective approach for reducing the expense of biocatalysts. It is a simple method that eliminates the need for additional purification and immobilization steps. Certain enzymes have been effectively exhibited on the cell surface as whole-cell biocatalysts, making them the ideal candidates for industrial biotransformation processes [18]. However, compared to other enzymes, the use of PLD in the form of cell-surface display is less common [12,14,19].

The super-folder green fluorescent protein (sfGFP) exhibits fluorescence, allowing direct observation without the need for equipment. Thus, by fusing the sfGFP tag to the target protein for expression, we can detect the presence of the secreted sfGFP fusion protein in the cell supernatant or culture medium through fluorescence observation [20]. The sfGFP expression system enables the rapid screening of high-level expression strains through fluorescence detection and facilitates the secretion of target proteins [21,22]. To enhance the efficiency of PLD preparation, we developed an sfGFP-based system in *Escherichia coli* for expressing and displaying PLD. This system enables the surface display of PLD on the bacteria. By utilizing the displayed sfGFP-PLDr34, we demonstrated that under optimal conditions, almost all PC was converted into PS. The surface display not only allows for the direct whole-cell preparation of PS but also facilitates the high-level screening of target proteins.

## 2. Materials and Methods

### 2.1. Plasmids, Strains, and Chemical Reagents

The laboratory stored *Escherichia coli* (*E. coli*) Rosetta Blue (DE3) and DH5α. The expression vectors, pET28a, pET28a-sfGFP, and pET23a-sfGFP, were maintained in our laboratory. Genecreate Co. (Wuhan, China) synthesized the plasmid pET28a-PLDr34 encoding PLDr34. This laboratory prepared competent *E. coli* DH5α and *E. coli* Rosetta Blue (DE3). Avanti Polar-Lipids, Inc. (Alabaster, AL, USA) provided PC (purity 95%, from soybean), and Sigma-Aldrich Co. (St. Louis, MO, USA) provided PS standards. Solarbio (Beijing, China) provided L-serine and peroxidase, while Beyotime (Shanghai, China) provided choline oxidase. Sangon Co. (Shanghai, China) synthesized the primers (Appendix A).

### 2.2. Construction of Recombinant Plasmids for Displaying Proteins on the Cell Surface

The PLDr34 (GenBank accession number MN604233) was synthesized and cloned into the vectors pET28a, pET28a-sfGFP, and pET23a-sfGFP downstream from the 6× His tag. The sfGFP and PLDr34 are connected by a 3C site, as shown in Appendix A. The recombinant plasmids were confirmed by sequencing.

### 2.3. Preparation of Whole-Cell Target Protein

The plasmids and strains utilized in this study are listed in Appendix A. The recombinant plasmids pET28a-PLDr34, pET28a-sfGFP-PLDr34, and pET23a-sfGFP-PLDr34 were transformed into *E. coli* Rosetta Blue (DE3) competent cells. One of the positive clones was inoculated into 5 mL of LB medium supplemented with 50 μg/mL of kanamycin and incubated at 220 rpm and 37 °C. Subsequently, the overnight cultures were transferred to 100 mL of LB medium containing 50 μg/mL of kanamycin and incubated at 220 rpm and 37 °C. Once the OD_600_ reached 0.6–0.8, the cells were induced with IPTG at a final concentration of 1 mM and incubated at 220 rpm and 18 °C for 18 h. Then, the cells were harvested by centrifugation at 5000× *g* for 5 min at 4 °C. The cells were accurately weighed, and an appropriate volume of phosphate-buffered saline (PBS) with a pH of 7.4 was added to achieve a cell concentration of 100 mg/mL. The resuspended cells could then be directly utilized for enzyme activity assays.

### 2.4. Cell Fractionation

Cell fractionation was performed according to the protocol described by Quan et al. [23]. Cells were harvested from a 10 mL culture broth by centrifugation at 5000× *g* for 10 min at 4 °C. The cells were then treated with one-third volume of Tris-EDTA-NaCl solution (TEN) containing 50 mM Tris-HCl (pH 8.0), 5 mM EDTA, and 50 mM NaCl. The mixture was incubated at 4 °C overnight. After incubation, the solution was centrifuged at 5000× *g* for 20 min at 4 °C. The supernatant was collected as the outer membrane fraction.

### 2.5. Fluorescence Microscopy

Cells were harvested from 1 mL of culture by centrifugation at 3000× *g* for 5 min at 4 °C. The harvested cells were then resuspended in 1 mL of phosphate buffer. The resuspended cells were diluted with phosphate buffer to the desired concentrations. Subsequently, 10 µL of the diluted cells was pipetted onto Poly-prep microscopy slides. Using a Zeiss LSM 980 fluorescence microscope (Zeiss, Oberkochen, Germany), the magnification of the objective lens was adjusted to locate the cells of interest. The LSCM was configured to scanning mode, the laser intensity parameters were fine-tuned, and ZEN v3.0 software was utilized to capture crisp confocal images.

### 2.6. Verification of PLD and sfGFP-PLD through Western Blot Analysis

The 10 μL of whole cells, outer membrane fraction, and medium supernatant from both pET28a-PLDr34 and pET28a-sfGFP-PLDr34 cells were employed for Western blot verification. Mouse-derived antibodies were used to bind the 6× His tagged-PLD and 6× His tagged-sfGFP-PLD proteins, and goat-derived antibodies were used to bind the mouse antibodies. The PVDF membrane was uniformly coated with the SuperKineTM Ultrasensitive ECL Luminescent Solution, followed by imaging using an ultra-sensitive chemiluminescence detector.

### 2.7. PC Hydrolysis Activity Assay

The hydrolysis activity of PLDr34 or sfGFP-PLDr34 (in the form of outer membrane fraction or intact cells) was detected using an enzyme-linked colorimetric assay [24]. The 200 μL reaction mixture consisted of 0.1% Triton X-100, 40 mM Tris–HCl (pH 7.5), 10 mM CaCl_2_, and 10 mg/mL PC dissolved in an 80% ethanol solution. The mixture was incubated at 37 °C and 200 rpm for 5 min. Subsequently, 200 μL of resuspended cells (100 mg/mL) was introduced into the reaction system, and the final reaction system was 400 μL. The reaction system was incubated at 37 °C and 200 rpm for a duration of 10 min. To terminate the reaction, 200 μL of a reaction termination solution (0.1 M Tris-HCl, 10 mM EDTA, and 10 g/L Hexadecyl trimethyl ammonium chloride) was added and incubated at 37 °C and 200 rpm for 5 min. The resulting mixture was centrifuged at 12,000 rpm for 5 min, and the supernatant was added to 2.5 mL of a chromogenic solution containing 1 mg of phenol, 0.3 mg/mL of 4-Aminoantipyrine, and 0.1 M Tris-HCl (pH 8.0). Additionally, 2 units of choline oxidase and 2 units of catalase were added to the mixture. The mixture was then incubated at 37 °C and 200 rpm for 30 min. The absorbance of the reaction mixture was measured at 500 nm. One unit (U) of hydrolytic activity of PLD was defined as the amount of enzyme that produced 1 µmol of choline per minute. The calibration curve was generated using a standard solution of choline chloride rather than the enzyme solution.

### 2.8. PS Synthesis Catalyzed by PLD

The transphosphatidylation of PC and L-serine to PS was carried out in a two-phase reaction system as described by Haiyang Zhang et al. [22]. Under the optimized reaction conditions, the final concentration of whole cells was 10 mg/mL. Therefore, 200 μL of 100 mg/mL resuspended cells was centrifuged at 12,000 rpm for 3 min. The supernatants were discarded, and the cell pellets were resuspended with 1.0 mL of a sodium acetate/acetic acid buffer (0.02 M, pH 6.0) containing serine (1.0 M) and CaCl_2_ (0.1 M). Then, 1.0 mL of PC (20 mg/mL) dissolved in diethyl ether was introduced into the reaction system and incubated at 45 °C for 4 h. Following that step, 2 μL aliquots of the sample and standard solutions were applied to the silica gel plate. A mixture of chloroform, methanol, glacial acetic acid, acetone, and water (45:25:7:4:2) served as the developing agent. The thin-layer plate was removed, and the solvent was allowed to evaporate. Subsequently, the plate was placed in an iodine cylinder containing a mixture of iodine and silica gel powder for 2 min to observe color development. The conversion of PS was calculated as the ratio of generated PS to the sum of generated PS and unreacted PC.

### 2.9. Treatment of Cells by Proteinase K

The cell suspension, resuspended in 200 μL of PBS, was divided into two tubes. One tube received 10 μL of protease K (20 mg/mL), and both tubes were incubated at 37 °C for 1 h. The supernatant was removed via centrifugation at 5000× *g* for 10 min. The precipitate was washed three times with PBS and then resuspended in PBS to assess its hydrolysis activity.

### 2.10. Determination of Cell Growth Curve

To compare the growth status of pET28a-sfGFP-PLDr34 and pET23a-PLDr34 cells, the recombinants pET28a-PLDr34 and pET28a-sfGFP-PLDr34 were transformed into *E. coli* Rosetta Blue (DE3) competent cells. The colonies were picked and cultured in 5 mL LB medium with 50 μg/mL kanamycin at 220 rpm and 37 °C. Once the OD_600_ of the two strains reached 0.8, they were diluted with LK medium (LB broth supplemented with 50 μg/mL kanamycin). Subsequently, 10 μL aliquots of each strain were inoculated into fresh 5 mL LK medium. The cultures were then incubated at 37 °C with shaking at 220 rpm. Samples were collected at a frequency of every 2 h for the initial 14 h, followed by every 4 h for the subsequent 14 h. The growth curve was obtained by measuring the OD_600_ value. IPTG was added at a concentration of 1 mM when the OD_600_ value of both bacteria reached 0.6. The cultures were incubated at 220 rpm and 18 °C.

## 3. Results

### 3.1. Expression of the Target Proteins

The PLDr34 gene was synthesized and cloned into the vectors pET28a and pET28a-sfGFP, as illustrated in Figure 1a. Recombinant strains *E. coli* Rosetta Blue (DE3)/PLDr34 and *E. coli* Rosetta Blue (DE3)/sfGFP-PLDr34 were cultured following the methods described in the Materials and Methods section. The total cellular proteins were analyzed using SDS-PAGE. As depicted in Appendix A, although PLDr34 (~60 kDa) and sfGFP-PLDr34 (~86 kDa) were detected in the total cells, their expression levels were not high.

To confirm the successful display of sfGFP-PLDr34 on the cell surface, we analyzed the outer membrane of the cells using a Western blot. As depicted in Figure 1b, sfGFP-PLDr34 and PLDr34 exhibit comparable expression levels, with both being relatively low. sfGFP-PLDr34 was predominantly present in the outer membrane of the cells, although it was also detected in the supernatant of the culture medium. PLDr34 was not detected in either the outer membrane of cells or the supernatant of the culture medium. Moreover, we used LSCM to detect the distribution of sfGFP-PLDr34 in *E. coli*. Figure 1c shows the obtained images from scanning in both bright and dark fields of view. The results revealed that sfGFP-PLDr34 bacterial cells displayed intense spontaneous fluorescence, with sfGFP also present in the background. These results indicate that sfGFP effectively promotes the display of PLD on the outer membrane of *E. coli*.

### 3.2. Surface Display of Fusion Enzymes on E. coli for PS Biosynthesis

We examined the hydrolytic activity of whole cells containing PLD. The strains carrying pET28a-PLDr34 and pET28a-sfGFP-PLDr34 were treated with proteinase K. Figure 2a shows that pET28a-PLDr34 cells without proteinase K treatment exhibited an enzyme activity of approximately 0.4 U/mL. Upon treatment with proteinase K, the activity decreased to around 0.2 U/mL, indicating that PLDr34 is active to some extent outside the cells. This finding is consistent with previous reports [12]. The constructed pET28a-sfGFP-PLDr34 cells showed the highest activity (~1 U/mL), which was 2.5 times higher than the pET28a PLDr34 strain and 10 times higher than the pET28a-sfGFP-PLDr34 cells treated with protease K. This observation confirms the successful display of sfGFP-PLDr34 on the cell surface of *E. coli*.

Then, we measured the ability of *E. coli* Rosetta Blue (DE3)/pET28a-PLDr34 whole cells and *E. coli* Rosetta Blue (DE3)/pET28a-sfGFP-PLDr34 whole cells to convert PC into PS. After a reaction at 40 °C for 2 h, the TLC results showed that sfGFP-PLDr34 cells produced PS, while PLDr34 did not produce PS (Figure 2b). In order to further improve the efficiency of PS production, we also tested the activity of the pET23a sfGFP-PLDr34 strain, because the regulation of pET23a is more relaxed than pET28a. However, its activity is lower than pET28a-sfGFP-PLDr34 cells (Figure 2b). We speculate that due to the toxicity of PLD to cells [13,17], strict regulation of the pET28a strain can reduce the impact of PLD expression on cell growth, and sfGFP-mediated secretion expression further reduces the cytotoxicity of PLD (Appendix A).

### 3.3. Effects of Cell Concentration

The reaction is significantly influenced by the concentration of cells. In order to examine this effect, we conducted a PS conversion reaction using sfGFP-PLDr34 cells at concentrations ranging from 0.25 mg/mL to 25 mg/mL. To prepare reaction mixtures containing cells at different concentrations, we aspirated varying volumes of cells resuspended in PBS buffer at a concentration of 100 mg/mL. After centrifuging to remove the supernatant, the cell pellets were resuspended with the reaction mixture.

Figure 3a demonstrates a strong correlation between the cell concentration and the yield of PS. PS begins to appear at a cell concentration of 2.5 mg/mL. PC completely disappears at a cell concentration of 10 mg/mL. But as the cell concentration further increases, PS also decreases. Therefore, considering the cell cost and PS conversion rate, 10 mg/mL was chosen as the optimal cell concentration. We also tested the effect of cell concentration on hydrolysis activity. As shown in Figure 3b, the optimal cell concentration for hydrolysis activity is also 5–10 mg/mL. A further increase in cell concentration will lead to a significant decrease in hydrolysis activity. We speculate that cells themselves might metabolize a portion of PC, and more cells metabolize more PC, resulting in a decrease in PS production and PC hydrolysis activity with a higher cell concentration. Therefore, it is necessary to choose a balance point between cell concentration and activity.

### 3.4. Effects of pH and Ca^2+^

To determine the effect of pH on the reaction, we evaluated the PS conversion activity and hydrolysis activity of sfGFP-PLDr34 cells at different pH levels. As shown in Figure 4a, the optimum pH for the PS conversion reaction was 5.0–6.0. Interestingly, the sfGFP-PLDr34 cells showed optimal hydrolysis activity at pH 5.0 (Figure 4b). However, in previous reports, purified free PLDr34 exhibited the highest hydrolysis activity at pH 6.0 [12]. We hypothesize that the shift in optimal pH could be attributed to alterations in the microenvironment surrounding the enzyme when displayed on the cell surface. This phenomenon has also been observed in a study where PLD was displayed on the surface of *Pichia pastoris* [19].

The activity of PLD requires divalent metal ions, which are usually most active in the presence of Ca^2+^ [14]. We evaluated the PS conversion activity and hydrolysis activity of sfGFP-PLDr34 cells at different Ca^2+^ concentrations. Surprisingly, there is a significant difference in the optimal Ca^2+^ concentration between PS synthesis activity and PC degradation activity. As shown in Figure 4a,b, the optimum Ca^2+^ concentrations for the PS conversion activity and hydrolysis activity are 100 mM and 10 mM, respectively. Although no PS was detected without the addition of Ca^2+^, weak hydrolytic activity could still be detected. Previous studies typically reduced the hydrolysis activity of PC and improved the synthesis activity of PS by engineering PLD [12]. Our results suggest that by adjusting the Ca^2+^ concentration, PLD can maintain low hydrolysis activity while maintaining high PS synthesis activity, thereby improving the PS conversion efficiency.

### 3.5. Effects of Reaction Temperature

We also assessed the impact of reaction temperature on the PS conversion activity. Figure 5a demonstrates that at 45 °C and 50 °C, no PC was detected, only PS was observed. Due to the higher abundance of PS products at 45 °C, this temperature was chosen as the optimal reaction temperature for sfGFP-PLDr34 cells.

Subsequently, we examined the reaction time required to generate PS. The formation of the PS product was observed after a 1 h reaction at 45 °C, as depicted in Figure 5b. After 2 h, the majority of PC had been converted into PS. By the end of the 4 h reaction, PC had almost completely disappeared, and the amount of PS reached the maximum. Hence, we established the reaction conditions for the majority of the reactions at 45 °C for a duration of 4 h.

### 3.6. Effects of the Concentration of PC and L-Serine

We also examined the impact of substrate concentration on the reaction. The PS yield increased as the PC concentration increased when the concentration of L-serine was 1 M (Figure 6a). When the PC concentration was ≤20 mg/mL, almost all of it could be converted. Nevertheless, a further increase in the PC concentration could not be fully converted. Importantly, when we increased the concentration of L-serine to 2 M, and even with an increased concentration of PC to 50 mg/mL, most of the PC could still be converted to PS. This finding suggests that increasing the concentration of L-serine effectively promotes the PS conversion activity, which aligns with previous reports [14].

### 3.7. Reuse of Whole-Cell Catalyst

High operational stability is crucial for most biological processes and can reduce the cost of biotransformation in commercial applications. We evaluated the stability of sfGFP-PLDr34 cells over five consecutive 4 h batches. The PS conversion activity decreased significantly with each reuse. As indicated in Figure 7a, the PS yield remained at approximately 40% and 10% in the second and third cycles, respectively. It was not detected in the fifth cycle. From a practical standpoint, sfGFP-PLDr34 could be reused for a maximum of four times. Nevertheless, this number of reuses is comparable to the previously reported results of displaying PLD on the surface of *E. coli* using the autotransporter domain of AIDA-I [12].

The fusion protein with an sfGFP tag can be directly visible without the need for any equipment [21]. This provides a significant advantage over other tags. Furthermore, the fusion enzyme sfGFP-PLDr34 can be expressed on the surface of *E. coli* cells, enabling intact cells to exhibit green fluorescence. As shown in Figure 7b, the green fluorescence of intact cells gradually decreased after each reaction. When the fluorescence became very weak or invisible, it indicated that the intact cells were no longer viable. Therefore, we can determine whether to terminate the recycling reaction based on the fluorescence intensity, without the need for enzyme activity detection. This approach is highly convenient and cost effective, especially for large-scale industrial applications.

## 4. Discussion

By utilizing sfGFP to display PLD on the cell surface, we have developed a highly efficient, visible, simple, and cost-effective method to produce PS from PC and L-serine using the whole-cell sfGFP-PLDr34 catalyst. This approach is appealing because it eliminates the requirement for additional steps of enzyme extraction and purification. Furthermore, the visible fusion of sfGFP-PLDr34 immobilized on the bacterial surface is relatively stable under reaction conditions. Hence, the reaction conditions can be easily controlled, particularly for large-scale industrial applications. The entire cell can be reused up to three rounds while still producing detectable levels of PS. Most importantly, this system can be used for the rapid screening of higher-activity PLDs, as the fluorescence of sfGFP can indicate the expression level of the fused PLD and its changes during repeated use. Therefore, the use of sfGFP-mediated PLD displayed cells as a biocatalyst for cost-effective PS production shows promise.

However, we also acknowledge several limitations in this study. Firstly, the *E. coli* Rosetta Blue (DE3) strain we employed produces endotoxin, and this would be incompatible with food, cosmetics, and pharmaceutical usage, thus enormously limiting this study. To adapt this method for these applications, alternative probiotic strains such as *E. coli* Nissle 1917 (EcN) [25] could be explored for sfGFP-PLD display. Secondly, the reuse times of whole cells need to be improved, which may be achieved through three strategies. (1) We observed that the expression level of sfGFP-PLD is relatively low. This can be improved by optimizing expression conditions, such as testing various inducer concentrations and induction temperatures. (2) Alternative PLDs with higher stability and activity could be selected for surface display, such as the rationally designed *Sr*_MBP_PLD^Mu6^ by Qi et al. [26]. (3) Immobilization can enhance the stability of the catalyst. Recently, various strategies have been reported to improve PLD performance through immobilization [15,27,28,29,30,31,32,33]. We could explore the use of immobilization reagents to immobilize the cells or elute the membrane-bound sfGFP-PLD for enzyme immobilization. We believe that with the resolution of these limitations, sfGFP-based PLD display technology will significantly promote the biosynthesis of phospholipids, including PS.

## 5. Conclusions

In summary, our study has successfully achieved the efficient extracellular production of PS using whole-cell display technology in *E***.**
*coli*, providing a novel pathway for the biosynthetic production of PS. This approach is appealing as it eliminates the additional steps of enzyme extraction and purification. Furthermore, this system facilitates rapid screening of PLDs exhibiting higher activity. The fluorescence intensity of sfGFP serves as a proxy for the expression level of the fused PLD, allowing for the detection of changes during repeated use. In the future, we aim to further investigate and enhance the production efficiency of PS while reducing costs, thereby contributing to the advancement of related industries.

## Figures and Tables

**Figure 1 biomolecules-14-00430-f001:**
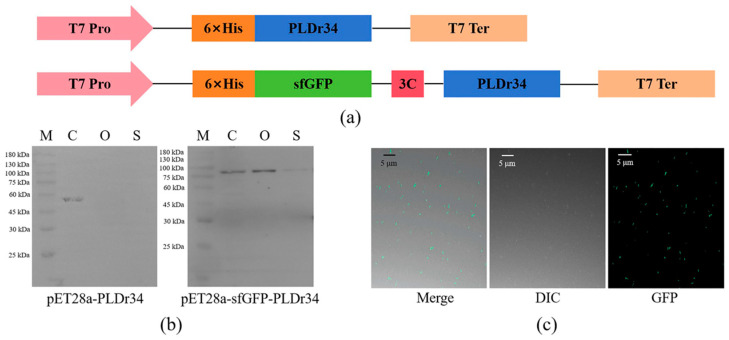
Construction of PLDr34 cell-surface display system for recombinant plasmids and analysis of PLDr34 and sfGFP-PLDr34. (**a**) Construction of recombinant plasmid. T7 Pro: T7 promoter; 3C: 3C site; T7 Ter: T7 terminator. (**b**) Western blot analysis of whole cells, washing membrane, and culture medium protein. M: protein marker; C: whole-cell fraction; O: washing membrane fraction; S: culture medium fraction. Original image can be found in Appendix A Appendix A. (**c**) LSCM was used to detect the distribution of the sfGFP-PLDr34 protein in *E. coli*. The images obtained by scanning in merge, bright, and dark fields are represented by Merge, DIC, and GFP, respectively.

**Figure 2 biomolecules-14-00430-f002:**
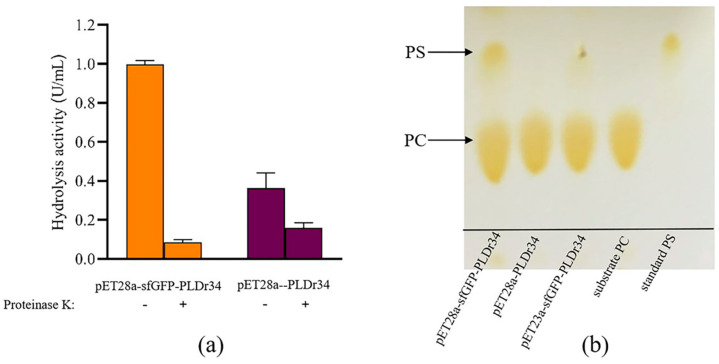
Exploration of whole-cell activity. (**a**) Demonstrating the hydrolytic activity of. PLDr34 in full cells. Control groups of pET28a-PLDr34 were divided into those not treated and those treated with proteinase K. The surface display group of pET28a-sfGFP-PLDr34 was also divided into those not treated and those treated with proteinase K. The data were derived from the mean and standard deviation of three independent experiments. (**b**) Comparing the whole-cell catalysis of PC synthesis to PS.

**Figure 3 biomolecules-14-00430-f003:**
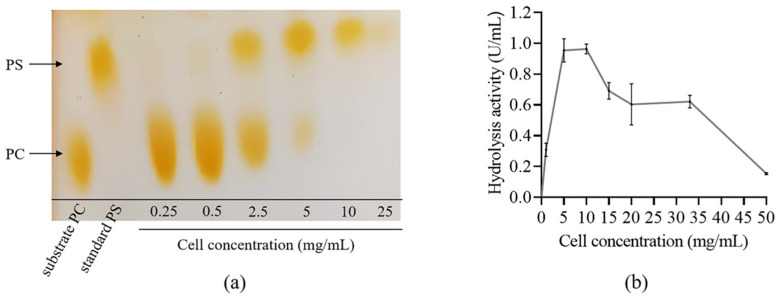
Effects of cell concentration. (**a**) The impact of cell concentration on PS synthesis activity. (**b**) The influence of cell concentration on hydrolysis activity. The data were derived from the mean and standard deviation of three independent experiments.

**Figure 4 biomolecules-14-00430-f004:**
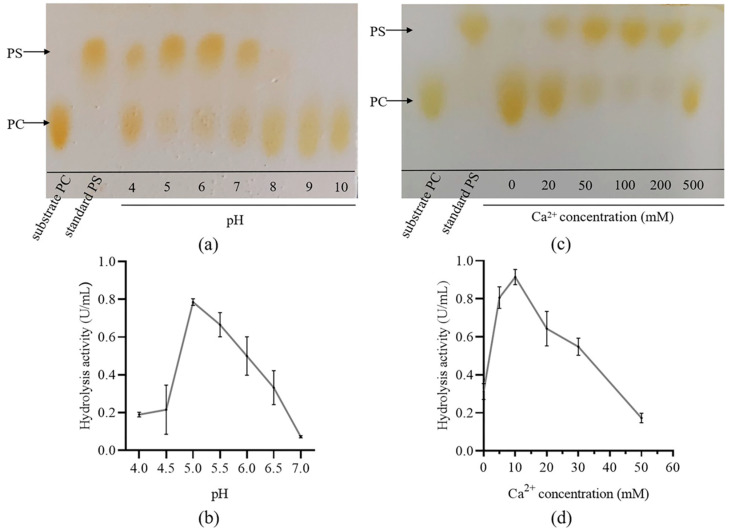
Effects of pH and Ca^2+^. (**a**) The impact of pH on PS synthesis activity. (**b**) The effects of pH on enzyme hydrolysis activity. (**c**) The effects of Ca^2+^ concentration on the activity of PS synthesis. (**d**) The effects of Ca^2+^ concentration on enzyme hydrolysis activity. The data in (**b**,**d**) were derived from the mean and standard deviation of three independent experiments.

**Figure 5 biomolecules-14-00430-f005:**
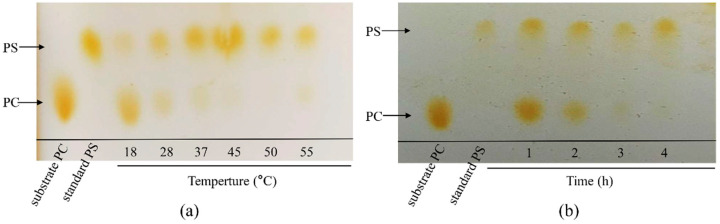
The impact of temperature and time gradient on the synthesis of PS. (**a**) The influence of temperature on the efficiency of PS synthesis. (**b**) The outcomes of varying time gradients in the catalytic synthesis of PS from PC.

**Figure 6 biomolecules-14-00430-f006:**
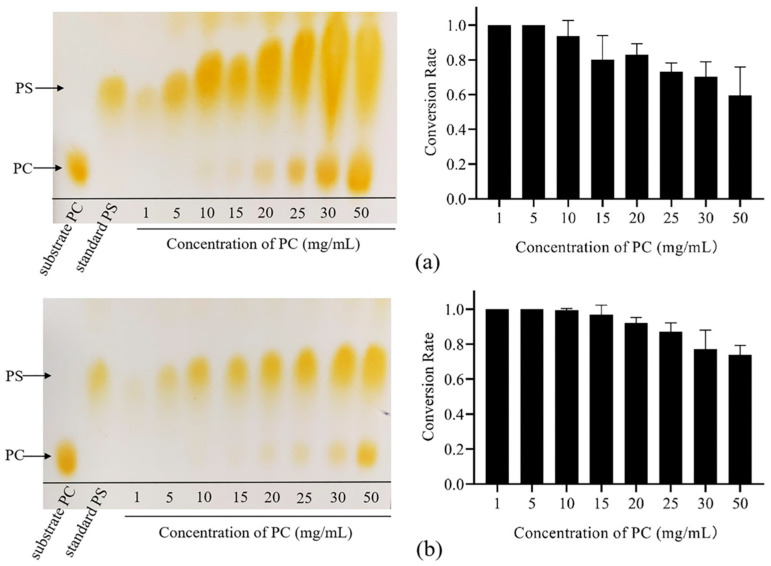
Effects of the concentration of PC and L-serine. (**a**) The results of PS synthesis at different PC concentrations with 1 M L-serine. (**b**) The results of PS synthesis at different PC concentrations with 2 M L-serine. The data were derived from the mean and standard deviation of three independent experiments.

**Figure 7 biomolecules-14-00430-f007:**
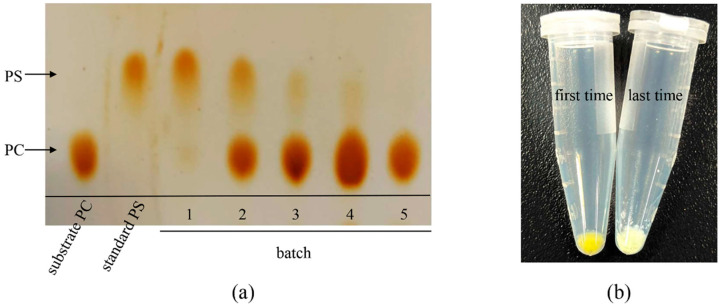
Reuse of whole-cell catalyst. (**a**) TLC results of whole-cell cyclic reaction catalyzing the production of PS. (**b**) Comparison of whole-cell fluorescence after five repetitions of pET28a-sfGFP-PLDr34 reuse.

## Data Availability

All relevant data of this study are presented. Additional data will be provided upon request.

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
