# Peer review of "Whole-Cell Display of Phospholipase D in Escherichia coli for High-Efficiency Extracellular Phosphatidylserine Production"

_biomolecules, 2024, doi:10.3390/biom14040430_

Round 1
Reviewer 1 Report
Comments and Suggestions for Authors
The authors describe a method of whole cell catalysis by phospholipase D in E. coli. This reviewer recommends publications after a few minor revisions.
Comments to address:
1. General comment: please be specific about the temperature of the reactions being monitored.
2. Figure 1: the letters need to be more explicit, either larger or in the right spot so they are unambiguous.
3. Figure 2: check the spelling of proteinase K.
4. Figure 7: a and b are reversed.
5. References are a little sparse with very few in the 2020’s. Please add more references that are timely.
Author Response
1. General comment: please be specific about the temperature of the reactions being monitored.
Response: We thank the referee for pointing this. We have indicated the reaction temperature in Materials and Methods.
2. Figure 1: the letters need to be more explicit, either larger or in the right spot so they are unambiguous.
Response: We thank the referee for pointing this. We have modified the letters.
3. Figure 2: check the spelling of proteinase K.
Response: We thank the referee for pointing out this error. We have modified this error.
4. Figure 7: a and b are reversed.
Response: We thank the referee for pointing out this error. We have modified this error.
5. References are a little sparse with very few in the 2020’s. Please add more references that are timely.
Response: We thank the referee for pointing this. We have added more references that are timely.
Reviewer 2 Report
Comments and Suggestions for Authors
In eukaryotic cells, phosphatidylserine (PS) is an important plasma membrane component that plays a significant role in programmed cell death. PS is also essential in neuronal membranes and myelin sheaths and is a global immune suppressor controlling phagocytosis and inflammation. Thus, it is widely used in the pharmaceutical, cosmetic, and food industries. Nowadays, the most popular approach to producing PS is to extract it from biomass, egg yolk, soybeans and vegetable oil. However, this approach foresees quite high costs.
Therefore, alternative approaches are currently being evaluated. One of these is to synthesize PS enzymatically, by exploiting a transphosphatidylation reaction, guided by the Phospholipase D (PLD) enzyme, starting from Phosphatidylcholine and L-Serine.
The authors of the study titled "Whole-Cell Display of Phospholipase D in Escherichia coli for High-Efficiency Extracellular Phosphatidylserine Production" report the development of a system, based on the super-folder Green Fluorescent Protein (sfGFP), in E. coli to produce a soluble and extracellular form of PLD. Afterward, they determined the optimal conditions for the higher yield by taking benefit of the "tools" generated, namely the engineered E. coli strains overexpressing the chimeric protein sfGFP-PLD.
Overall, though a very similar inquiry was carried out a not long time ago by using Streptomyces as a host system (https://doi.org/10.1021/acs.jafc.9b05394), the subject attracted the interest of the reviewer. Nonetheless, before publication, there are quite a few issues that must be properly addressed.
First of all, since the protein is His tagged a Western blot instead of a Coomassie-stained gel is required to assess the exogenous protein expression and its yield.
Second, from the manuscript, it is not clear whether the exogenous protein is membrane-bound or secreted, because sometimes, throughout the main text, the authors state that is secreted (i.e., in the section "Abstract") and then exposed on the cell surface (i.e., section 3.2. "…on the cell surface of E. coli…"). This is a little bit confusing for the readers. Additionally, the authors are warmly invited to provide experimental evidence for that (e.g., checking the presence of protein in the supernatant by immunoprecipitation).
Figures 3A and 3B: a relevant discrepancy between the TLC and the quantification plot occurs. Additionally, how cell concentration has been determined? What does stand mg/mL? Protein? Bacteria? This should be sorted out.
Figures 4A and 4B: there are quite a few incongruencies that require to be addressed properly. a) 4A looking at the error bars three points (i.e., pH4, pH5 and pH7) in which the measurements do not display variability. This is quite odd if one compares the size of the error bars of the other points. This requires clarification; b) 4B the x-axis of the plot and TLC data do not fit (i.e., in TLC the Ca2+ concentration increases up to 500 mM whereas in the plot it stops at 50 mM).
The section "Material and Methods" needs implementation with some details (e.g., cloning sites? Solvent used for TLC? The concentration of Triton X-100 I guess that should be indicated as %, etc…)
The "Discussion" section is missing. This is a very important point the approach used by the authors, namely expressing E. coli bacteria might offer some technical advantages but it is also well known that sometimes bacteria produce endotoxins and this would be incompatible with food, cosmetics, and pharmaceutical usage, thus limiting enormously their study.
Minor issues
-Several typos are scattered throughout the main text (e.g., genus and species should be in italics; in Figure 1B what is the difference between the black and white arrows?).
Figure 1C: the authors are asked to provide a picture with a better definition because the fluorescence is barely visible and DIC is not visible at all.
Comments on the Quality of English LanguageThe English language would benefit from moderate editing throughout the main text.
Author Response
1. First of all, since the protein is His tagged a Western blot instead of a Coomassie-stained gel is required to assess the exogenous protein expression and its yield.
Response: Thanks a lot for your suggestions. We conducted a Western Blot experiment (Fgure 1b), which demonstrated successful expression of both sfGFP-PLDr34 and PLDr34, albeit at low levels. We have rephrased the corresponding results according to the new results as follows: “To confirm the successful display of PLDr34 and sfGFP-PLDr34 on the cell surface, we analyzed the outer membrane of the cells using Western blot. As depicted in Figure 1b, sfGFP-PLDr34 and PLDr34 exhibit comparable expression levels, with both being relatively low.”
2. Second, from the manuscript, it is not clear whether the exogenous protein is membrane-bound or secreted, because sometimes, throughout the main text, the authors state that is secreted (i.e., in the section "Abstract") and then exposed on the cell surface (i.e., section 3.2. "…on the cell surface of E. coli…"). This is a little bit confusing for the readers. Additionally, the authors are warmly invited to provide experimental evidence for that (e.g., checking the presence of protein in the supernatant by immunoprecipitation).
Response: We thank the referee for pointing this. We analyzed the distribution of the protein using Western blot (Figure 1b), revealing that sfGFP-PLDr34 is primarily localized to the outer membrane of the cell, with a minor fraction distributed in the supernatant of the culture medium. This suggests that sfGFP-PLDr34 is primarily membrane-bound, with only a minor fraction of the protein being secreted into the culture medium. We have made corresponding modifications to the relevant sentences.
3. Figures 3A and 3B: a relevant discrepancy between the TLC and the quantification plot occurs. Additionally, how cell concentration has been determined? What does stand mg/mL? Protein? Bacteria? This should be sorted out.
Response: We thank the referee for pointing this. TLC analysis in Figure 3A demonstrates the activity of PS synthesis, while the quantitative representation in Figure 3B illustrates the activity of PC hydrolysis. These two activities exhibit distinct requirements for cell concentration. The mg/mL stands the bacteria. To detrmine the cell concentration, we determined the weight of the bacterial cells obtained by centrifugation at 5000× g for 5 min at 4 °C. Then, the cells were resuspended to a concentration of 100 mg/mL using phosphate-buffered saline (PBS) with a pH of 7.4. To prepare reaction mixtures containing cells at different concentration, we aspirated varying volumes of the resuspended cells, and centrifuged them at 12000 rpm for 3 minutes. Following supernatant removal, the cell pellet was resuspended in the reaction mixture. We have added these details to the “Materials and Methods” section and “Results” section.
4. Figures 4A and 4B: there are quite a few incongruencies that require to be addressed properly. a) 4A looking at the error bars three points (i.e., pH4, pH5 and pH7) in which the measurements do not display variability. This is quite odd if one compares the size of the error bars of the other points. This requires clarification; b) 4B the x-axis of the plot and TLC data do not fit (i.e., in TLC the Ca2+ concentration increases up to 500 mM whereas in the plot it stops at 50 mM).
Response: We thank the referee for pointing this. We have revised the depiction of Figure 4B to enhance the visibility of variability through the error bar. TLC analysis in Figure 4C demonstrates the activity of PS synthesis, while the quantitative representation in Figure 4D illustrates the activity of PC hydrolysis. These two activities exhibit distinct requirements for Ca2+ concentration. Thus the x-axis of the plot and TLC data do not fit.
5. The section "Material and Methods" needs implementation with some details (e.g., cloning sites? Solvent used for TLC? The concentration of Triton X-100 I guess that should be indicated as %, etc…)
Response: We thank the referee for pointing this. We have added these details in the “Materials and Methods” section.
6. The "Discussion" section is missing. This is a very important point the approach used by the authors, namely expressing E. coli bacteria might offer some technical advantages but it is also well known that sometimes bacteria produce endotoxins and this would be incompatible with food, cosmetics, and pharmaceutical usage, thus limiting enormously their study.
Response: We thank the referee for pointing this. We have added the Discussion section.
Minor issues
1. -Several typos are scattered throughout the main text (e.g., genus and species should be in italics; in Figure 1B what is the difference between the black and white arrows?).
Response: We thank the referee for pointing out these errors. We have modified these errors. Figure 1B has been relocated to Supplementary Figure S2, and all arrows in this figure have been changed to black.
2. Figure 1C: the authors are asked to provide a picture with a better definition because the fluorescence is barely visible and DIC is not visible at all.
Response: We thank the referee for pointing this. We have provided a picture with a better definition.
Reviewer 3 Report
Comments and Suggestions for Authors
In this manuscript, Sun et al., developed an efficient E-coli based system to produce and detect phosphatidylserine (PS) from phosphatidylcholine (PC). The authors thoroughly explored the effects of a range of conditions, including cell strains, cell concentration, pH, calcium concentration, temperature, time, and L-serine concentration on PS yielding. The reviewer only has a minor comment before its consideration of publication.
Figure 6: The effects of L-serine on PC yielding need a comparison with a better quantification. Could the authors normalize the PS/PC expression in Figure 6(a) and (b) to their loading controls (substrate PC and standard PS), respectively?
Author Response
Figure 6: The effects of L-serine on PC yielding need a comparison with a better quantification. Could the authors normalize the PS/PC expression in Figure 6(a) and (b) to their loading controls (substrate PC and standard PS), respectively?
Response: We thank the referee for pointing this. We have determined the conversion of PS by calculating the ratio of generated PS to the total amount of generated PS and unreacted PC. Figures 6(a) and 6(b) have been updated.
Reviewer 4 Report
Comments and Suggestions for Authors
In this paper, the authors present the expression of PLD on the surface of E. coli as a potentially industrially usable system for the production of phosphatidylserine. The paper is written in fairly good English, although there are parts that could be better worded. The chapter "Materials and methods" lacks descriptions of how some experiments were carried out (see remarks), and the biggest technical shortcoming of the work is that the authors did not prove the existence of this enzyme on the cell surface using the SDS-PAGE method. To publish the paper, they should prove that the protein is really on the cell surface using the western blot method. Furthermore, authors did not comment possible reason for diminishing of enzyme activity obtained with higher concentration of cells. Individual comments follow:
According to SDS-PAGE in FIg. 1 I could not say that any recombinant protein is expressed. Bends before and after induction are not different at all. Quality of (c) part of the figure is also very low - it is hard to see anything. Authors should show some western blot proof that the enzyme is present and made higher quality picture of fluorescence. Furthermore, it is not explained how did authors prepare samples for ˝whole cells˝ for SDS-PAGE? There is much more total protein in lines 3, 4, 7 and 8 than in other lines, so I don't think is correct to say that there is predominantly present sfGFP-PLD34 while PLDr34 is barely detected (lines 186-188).
lines 189 -192 (sentences from Adjust the magnification..... to ............confocal images.) are written as a protocol for microscopy, not as a part of scientific paper. This should be improved.
There is no protocol for proteinase K treatment of cells in the ˝Materials and Methods˝ chapter. This should be added to the manuscript.
How did authors explain lower PS production and lover hydrolytic activity with higher concentration of cells?
a) and b) part of the Figure 4. are replaced in the legend, and it is not shown that fluorescence is gradually decreased (as stated in the text).
In my opinion paper is generally interesting but it should be improved and corrected before publication.
Comments on the Quality of English LanguageThe paper is written in fairly good English, although there are parts that could be better worded. Minor corrections are needed.
Author Response
1. According to SDS-PAGE in FIg. 1 I could not say that any recombinant protein is expressed. Bends before and after induction are not different at all. Quality of (c) part of the figure is also very low - it is hard to see anything. Authors should show some western blot proof that the enzyme is present and made higher quality picture of fluorescence. Furthermore, it is not explained how did authors prepare samples for ˝whole cells˝ for SDS-PAGE? There is much more total protein in lines 3, 4, 7 and 8 than in other lines, so I don't think is correct to say that there is predominantly present sfGFP-PLD34 while PLDr34 is barely detected (lines 186-188).
Response: Thanks a lot for your suggestions. We conducted a Western Blot experiment (Fgure 1b), which demonstrated successful expression of both sfGFP-PLDr34 and PLDr34, albeit at low levels. The Western blot results showed that sfGFP-PLDr34 is primarily localized to the outer membrane of the cell, with a minor fraction distributed in the supernatant of the culture medium. This suggests that sfGFP-PLDr34 is primarily membrane-bound, with only a minor fraction of the protein being secreted into the culture medium. We have made corresponding modifications to the relevant sentences. Additionally, we made a higher quality picture of fluorescence (Figure 1c). We have made corresponding modifications to the relevant sentences.
2. lines 189 -192 (sentences from Adjust the magnification..... to ............confocal images.) are written as a protocol for microscopy, not as a part of scientific paper. This should be improved.
Response: Thanks a lot for your suggestions. We have rephrased these details and relocated to the Materials and Methods section.
3. There is no protocol for proteinase K treatment of cells in the ˝Materials and Methods˝ chapter. This should be added to the manuscript.
Response: Thanks a lot for your suggestions. We have added the protocol for proteinase K treatment of cells in the “Materials and Methods”.
4. How did authors explain lower PS production and lover hydrolytic activity with higher concentration of cells?
Response: We thank the referee for pointing this. We speculate that cells themselves might metabolize a portion of PC, and more cells metabolize more PC, resulting in a decrease in PS production and PC hydrolysis activity with higher cell concentration. Therefore, it is necessary to find a balance point between cell concentration and activity.
5. a) and b) part of the Figure 4. are replaced in the legend, and it is not shown that fluorescence is gradually decreased (as stated in the text).
Response: We thank the referee for pointing this. We have rephrared the text as follows: “As shown in Figure 4a, the optimum pH for PS conversion reaction 5.0-6.0. Interestingly, the sfGFP-PLDr34 cells showed the optimal hydrolysis activity at pH 5.0 (Figure 4b).”
Reviewer 5 Report
Comments and Suggestions for Authors
The manuscript submitted for review (ID biomolecules-2881088) is interesting and within the scope of the Biomolecules journal. I did not notice any major methodological or technical errors in the work. Its individual fragments are coherent and the goal is clearly formulated and achieved. However, I have some concerns about the results of the iThenticate report. I believe that before the work is officially accepted for printing, the authors should change some parts of the work so as to significantly reduce the iThenticate report values. In my opinion, the current level of similarity of this manuscript (40%) to other previously published works is too high. That's why I suggest small corrections that relate only to the issue of the iThenticate report.
Author Response
Response: We thank the referee for pointing this. We have revised some parts of the work to to align with the standard.
Round 2
Reviewer 2 Report
Comments and Suggestions for Authors
The revised version of the manuscript by Sung and colleagues has been improved when compared to the original one. I am grateful to the authors for the efforts made in clarifying the issues raised by the reviewer.
However, before publication, there are still a couple of gaps that require to be filled.
It would be greatly appreciated if the authors would discuss the discrepancy between the conditions determined for PS synthesis and those for PC hydrolysis because one, from the biochemical point of view, would expect they would be sequentially tightly linked.
In the version I have got Supplementary Materials section is missing. Please provide them.
Author Response
Response: Thanks a lot for the referee’s suggestions. After careful examination, we found that there were little discrepancies in the optimal cell concentration and pH for PS synthesis activity and PC hydrolysis activity, while there was a significant discrepancy in the optimal Ca2+ concentration. Therefore, in the revised manuscript, we have discussed the possibility of adjusting Ca2+ concentration to improve PS conversion efficiency. In addition, we have uploaded supplementary materials in this submission.
Reviewer 4 Report
Comments and Suggestions for Authors
Authors accepted all suggestions and improved the quality of presentation to satisfying level.
Author Response
Response: Thanks a lot for the referee’s suggestions.